

# Metaproteomics of saliva identifies human protein markers specific for individuals with periodontitis and dental caries compared to orally healthy controls

Daniel Belstrøm[1],[*], Rosa R. Jersie-Christensen[2],[*], David Lyon[3],
Christian Damgaard[1],[4], Lars J. Jensen[3], Palle Holmstrup[1] and
Jesper V. Olsen[2]

[1] Section of Periodontology and Microbiology, Department of Odontology, Faculty of Health and Medical Sciences, University of Copenhagen, Copenhagen, Denmark
[2] Proteomics Program, Novo Nordisk Foundation Center for Protein Research, Faculty of Health and Medical Sciences, University of Copenhagen, Copenhagen, Denmark
[3] Disease Systems Biology Program, Novo Nordisk Foundation Center for Protein Research, Faculty of Health and Medical Sciences, University of Copenhagen, Copenhagen, Denmark
[4] Institute for Inflammation Research, Center for Rheumatology and Spine Diseases, Rigshospitalet, Copenhagen University Hospital, Copenhagen, Denmark
[*] These authors contributed equally to this work.

Corresponding author
Daniel Belstrøm, dbel@sund.ku.dk

## ABSTRACT

**Background:** The composition of the salivary microbiota has been reported to differentiate between patients with periodontitis, dental caries and orally healthy individuals. To identify characteristics of diseased and healthy saliva we thus wanted to compare saliva metaproteomes from patients with periodontitis and dental caries to healthy individuals.

**Methods:** Stimulated saliva samples were collected from 10 patients with periodontitis, 10 patients with dental caries and 10 orally healthy individuals. The proteins in the saliva samples were subjected to denaturing buffer and digested enzymatically with LysC and trypsin. The resulting peptide mixtures were cleaned up by solid-phase extraction and separated online with 2 h gradients by nano-scale $C_{18}$ reversed-phase chromatography connected to a mass spectrometer through an electrospray source. The eluting peptides were analyzed on a tandem mass spectrometer operated in data-dependent acquisition mode.

**Results:** We identified a total of 35,664 unique peptides from 4,161 different proteins, of which 1,946 and 2,090 were of bacterial and human origin, respectively. The human protein profiles displayed significant overexpression of the complement system and inflammatory markers in periodontitis and dental caries compared to healthy controls. Bacterial proteome profiles and functional annotation were very similar in health and disease.

**Conclusions:** Overexpression of proteins related to the complement system and inflammation seems to correlate with oral disease status. Similar bacterial proteomes in healthy and diseased individuals suggests that the salivary microbiota predominantly thrives in a planktonic state expressing no disease-associated characteristics of metabolic activity.

## INTRODUCTION

Saliva is a biological fluid critically involved in maintenance of oral homeostasis (*Marsh et al., 2016*), as qualitative and quantitative changes of saliva associates with increased frequency and severity of diseases in the oral cavity (*Dawes et al., 2015*; *Almstahl & Wikstrom, 1999*). Furthermore, saliva is easily and non-invasively collected (*Giannobile et al., 2011*), making it interesting to screen for biomarkers associated with oral and general health and disease status (*Baum et al., 2011*; *Zhang et al., 2016*).

In the last decade, salivary biomarkers of periodontitis and dental caries have been intensively investigated (*Yoshizawa et al., 2013*; *Miller et al., 2010*). These include salivary bacterial profiles that differentiate in patients with periodontitis (*Paju et al., 2009*; *Belstrøm et al., 2014b*; *Belstrøm et al., 2016b*), dental caries (*Yang et al., 2012*; *Belstrøm et al., 2014a*; *Belstrøm et al., 2015*) and orally healthy individuals. Furthermore, increased salivary levels of inflammatory protein biomarkers such as interleukin-1$\beta$ (IL-1$\beta$), IL-6 and matrix metalloproteinase-8 (MMP-8) have been described to be associated with periodontal disease status (*Kinney et al., 2011*; *Ebersole et al., 2013*; *Rathnayake et al., 2013*; *Ebersole et al., 2015*). Recently, the salivary transcriptome has been assessed (*Spielman et al., 2012*), and some transcriptomic characteristics of saliva have been reported in patients with dental caries (*Do et al., 2015*). Collectively, these reports conclude that biomarkers of different biological origin may be adequately assessed in saliva samples and support the concept that the biological composition of saliva reflects individual oral health status.

Mass spectrometry-based proteomics enables characterization of the protein content in any sample, including proteins of human and bacterial origin. It thus provides the possibility for simultaneous characterization of bacterial and host specific differences of saliva associated with oral health and disease. Only three studies have so far attempted to perform metaproteomic analysis of saliva in oral health (*Rudney et al., 2010*; *Jagtap et al., 2012*; *Grassl et al., 2016*). To the best of our knowledge, no study has so far compared metaproteomic profiles of saliva from patients with periodontitis and dental caries to orally healthy individuals.

The aim of the present study was to characterize the salivary metaproteome in 30 saliva samples, and compare human and bacterial proteome profiles between patients with periodontitis, dental caries and orally healthy individuals. The hypothesis was that both bacterial and human subsets of salivary metaproteome would differentiate between individuals with different oral health status.

## MATERIALS AND METHODS

### Study population and sample collection

The study population, clinical examination and collection of saliva samples have been presented in detail (*Belstrøm et al., 2016b*). In brief, saliva production was induced by

chewing on a tasteless paraffin gum, and chewing-stimulated saliva samples were collected from 10 patients with periodontitis, 10 patients with dental caries and 10 orally healthy individuals following a standardized protocol (*Kongstad et al., 2013*). Immediately after collection saliva samples were divided into four aliquots and stored at −80 °C for further analysis. One aliquot has previously been analyzed by next-generation sequencing (the Human Oral Microbe Identification using Next Generation Sequencing, HOMI*NGS*) (*Belstrøm et al., 2016b*). All participants signed an informed consent prior to participation, and the study was approved by the regional ethical committee (H-15000856-53175) and reported to the Danish Data Authorization (2015-54-0970).

## Sample preparation

The saliva proteome samples were prepared as described in (*Jersie-Christensen, Sultan & Olsen, 2016*) with a few modifications. Briefly, 1 ml of saliva was mixed with 1.5 ml lysis buffer (9 M Guanidine hydrochloride, 10 mM Chloroacetamide, 5 mM *tris*(2-carboxyethyl)phosphine in 100 mM Tris pH 8.5) and heated for 10 min (99 °C) followed by 4 min of sonication.

Protein concentration was measured with Bradford protein assay and ranged from 1–2.5 mg/ml. All samples were digested with the same amount of Lysyl Endoproteinase (Wako, Osaka, Japan) in a ratio of 1:100 w/w calculated from the highest concentration for 2 h. Samples were diluted to a final volume of 10 ml with 25 mM Tris pH8 and digested overnight with Trypsin (modified sequencing grade; Sigma) in a 1:100 w/w ratio.

Digestion was quenched by adding 1 ml of 10% trifluoroacetic acid and centrifuged at 2,000 g for 5 min. The resulting soluble peptides in the supernatant were desalted and concentrated on Waters Sep-Pak reversed-phase $C_{18}$ cartridges (one per sample) and the tryptic peptide mixtures were eluted with 40% acetonitrile (ACN) followed by 60% ACN. Peptide concentrations were determined by NanoDrop (Thermo, Wilmington, DE, USA) measurement.

## Mass spectrometry analysis

A total of 1.5 μg peptide mixture from each sample was analyzed by online nano-scale liquid chromatography tandem mass spectrometry (LC-MS/MS) in turn. Peptides were separated on an in-house packed 50 cm capillary column with 1.9 μm Reprosil-Pur $C_{18}$ beads using an EASY-nLC 1000 system (Thermo Scientific). The column temperature was maintained at 50 °C using an integrated column oven (PRSO-V1; Sonation GmbH, Biberach, Germany). Buffer A consisted of 0.1% Formic acid, and buffer B of 80% ACN, 0.1% Formic acid. The flow rate of the gradient was 200 nl/min and started at 5% buffer B, going to 25% buffer B in 110 min, followed by a 25 min step going to 40% buffer B and continuing to 80% buffer B in 5 min for a 5 min wash and returning to 5% in 5 min and continuing for re-equilibration for 5 min.

The Q Exactive HF instrument (Thermo Scientific, Bremen, Germany) was run in a data dependent acquisition mode using a top 12 Higher-Collisional Dissociation (HCD)-MS/MS method with the following settings. Spray voltage was set to 2 kV, S-lens RF level at 50, and heated capillary at 275 °C. Full scan resolutions were set to

60,000 at m/z 200 and the scan target was $3 \times 10^6$ with a maximum fill time of 20 ms. Full-scan MS mass range was set to 300–1,750 and dynamic exclusion to 20 s. Target value for HCD-MS/MS scans was set at $1 \times 10^5$ with a resolution of 30,000 and a maximum fill time of 60 ms. Normalized collision energy was set at 28.

## Data analysis

All 30 raw LC-MS/MS data files were processed together using MaxQuant version 1.5.0.38 (*Cox & Mann, 2008*) with default settings and match between runs. The integrated Andromeda peptide search engine and a reversed database approach applying a 1% FDR at both peptide and protein level was used. The data was searched in two iterations analogous to a previously described metaproteomics database search strategy (*Jagtap et al., 2013*). First, the search space consisted of the full SwissProt protein database (*The UniProt Consortium, 2015*) and the Human Oral Microbiome database (*Chen et al., 2010*) (both downloaded August 2014). The resulting search output was then used for reduction of the search space after filtering on different parameters. As a quality control measure, proteins with less than two unique peptides were removed. Furthermore, we required proteins to be detected in at least five out of 30 samples. Accession numbers from the Majority protein IDs column in the proteinGroups.txt were used to retrieve information about Lowest Common Ancestor (LCA) for each protein group entry. To find the LCA of a protein group, accession numbers with the most peptide-associations were selected, mapped to species and their full taxonomic lineage. The lowest taxonomic rank of the intersection of the latter yielded the LCA. All LCA searches resulting in the parvorder Catarrhini (primates) were set to be human. LCAs at taxonomic rank of species and genera, as well as all of their descendants were used to create a new, reduced search space. The latter was used for the second iteration of MS data identification and quantification and all accession numbers within a protein group were used to perform LCA searches. The above functionality was achieved using the Python programming language. Species names from SwissProt and HOMD were mapped to NCBI taxonomic identifiers using UniProt (http://www.uniprot.org/docs/speclist) and NCBI resources (http://www.ncbi.nlm.nih.gov/Taxonomy/TaxIdentifier/tax_identifier.cgi), respectively. Full taxonomic lineages were retrieved from NCBI Taxonomy database dump files (ftp://ftp.ncbi.nlm.nih.gov/pub/taxonomy/). Taxonomic comparison at genus- and species level was performed using Mann-Whitney U test with Benjamini-Hochberg correction for multiple testing.

Protein intensities based on summed peptide MS signal intensities were quantile normalized using the limma package version 3.24.15 under R version 3.2.2. Only proteins identified with more than one peptide ("razor + unique") and present in more than five out of the 30 samples were considered for further analysis. The mass spectrometry proteomics data have been deposited to the ProteomeXchange Consortium via the PRIDE (*Vizcaíno et al., 2016*) partner repository with the dataset identifier PXD004319.

For comparative analysis of the human protein profiles, the normalized intensity values were log2 transformed. For principal component analysis (PCA), missing values were

replaced with the constant value 19, representing the lowest protein intensity value measured. Analysis of significance (ANOVA) between groups was performed with the software package Perseus (http://www.perseus-framework.org). The resulting differentially expressed proteins were clustered using Euclidian distance after scaling the data by subtracting the mean intensity value. All p-values were corrected for multiple comparisons.

### Functional annotation of bacterial proteins

Bacterial proteins from HOMD were searched against Hidden Markov Models (HMMs) of bacterial Nested Orthologous Groups (NOGs) from eggNOG (*Huerta-Cepas et al., 2016*) using HMMscan version 3.1 (http://hmmer.org) (*Eddy, 2009*). For each protein query the resulting hits were restricted by two criteria. E-values had to be equal or lower than 1e-4 and a maximum overlap of eight amino acids of HMMs was allowed (selecting hits with the lowest e-value). All corresponding NOG-names were used to retrieve Gene Ontology (GO)-terms as well as KEGG pathways from eggNOG.

### KEGG pathway enrichment and characterization

To gain insights into differences between the three sample groups, KEGG pathway enrichment was performed using a modified version of AGOtool (*Schölz et al., 2015*). Individual samples were grouped to sample categories and the three paired combinations used for the enrichment analysis. All bacterial protein groups with an LCA at rank genus or below were selected. Benjamini-Hochberg correction (FDR) of p-values was applied to correct for multiple testing. The FDR was set to 1%. The following additional filter criterion was applied. The fold change had to be equal or higher than 2 or equal or lower than 0.5.

To get a functional overview of the bacterial proteins, we characterized each individual sample group by counting the number of protein groups associated with each KEGG pathway. For visualization purposes (Fig. S2), we selected the most highly associated terms. Within each group the number of associations was converted to percentages, and the most highly associated terms retained, until a cumulative sum of 90% was reached. This reduced the number of KEGG terms from 135 to 50.

## RESULTS

### General findings

Biomass analyses based on summed protein intensity measures demonstrated that approximately 95 and 3% of the total protein biomass was of human and bacterial origin, respectively (Fig. 1). Food-related proteins and proteins that could not be assigned to kingdom level comprised the rest of the biomass. We identified a total of 35,664 unique peptides from 4,161 proteins, of which 97% of the identified proteins could be assigned as bacterial or human proteins, with almost equal numbers of the two (Fig. 1; Table 1). The protein biomass and numbers between groups showed no significant differences using ANOVA analysis (Fig. S1).

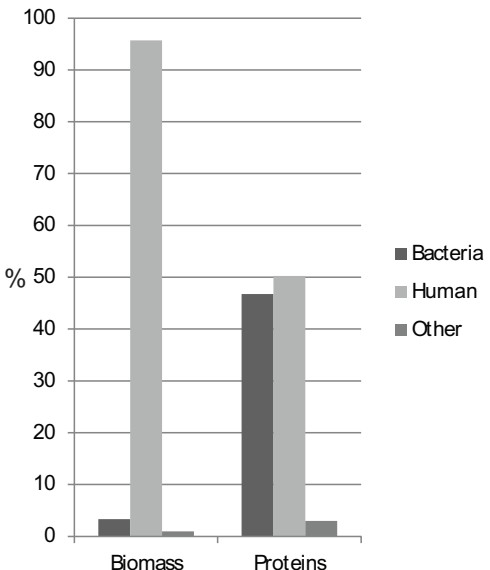

**Figure 1 Protein biomass and abundance across sample groups.** Relative distribution as a measure of summed intensity and protein count.

**Table 1 Overview of proteins identified.**

**Number of proteins**

| | Caries | Healthy | Periodontitis | Total* |
|---|---|---|---|---|
| Other** | 125 | 115 | 120 | 125 |
| Human | 2,084 | 2,079 | 2,084 | 2,090 |
| Bacteria | 1,861 | 1,926 | 1,924 | 1,946 |
| -mapped to genus level | 1,710 (91.9%) | 1,765 (91.6%) | 1,762 (91.6%) | 1,784 (91.7%) |
| -mapped to species level | 594 (31.9%) | 602 (34.1%) | 609 (34.6%) | 616 (34.5%) |
| Total | 4,070 | 4,120 | 4,128 | 4,161 |

**Notes:**
* Unique proteins.
** Food related proteins and proteins that could not be assigned to kingdom level.

## Human protein profiling

Principal component analysis (Fig. 2A) of the human proteins in saliva showed decent separation of samples from patients with periodontitis and dental caries from orally healthy individuals, based on the most decisive component of the dataset, accounting for 17.9% of the variation. The most enriched KEGG pathway in component 1 and 2 was 'Complement and coagulation cascades' (Fig. 2B). Component 2 also separated samples from patients with dental caries and periodontitis patients from orally healthy individuals with the component explaining 12.7% of the variation. Two of the most enriched terms in component 2 in the direction of the orally healthy individuals were KEGG pathway 'Salivary secretion' and GOBP 'protein glycosylation.'

From a total of 2,090 identified proteins of human origin, 60 proteins were significantly differentially expressed when performing multiple sample test (ANOVA, $p < 0.05$). Hierarchical cluster analysis of the proteins nicely separated the three sample groups,

A

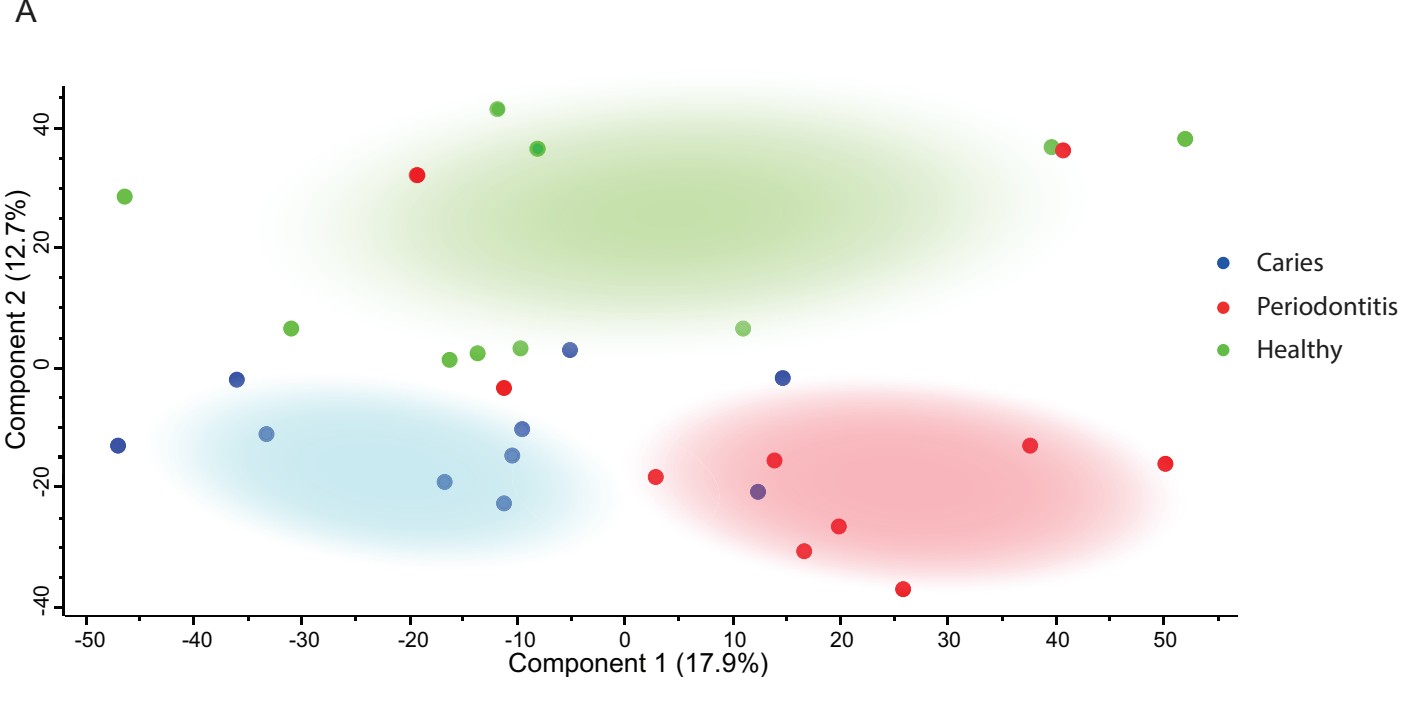

B

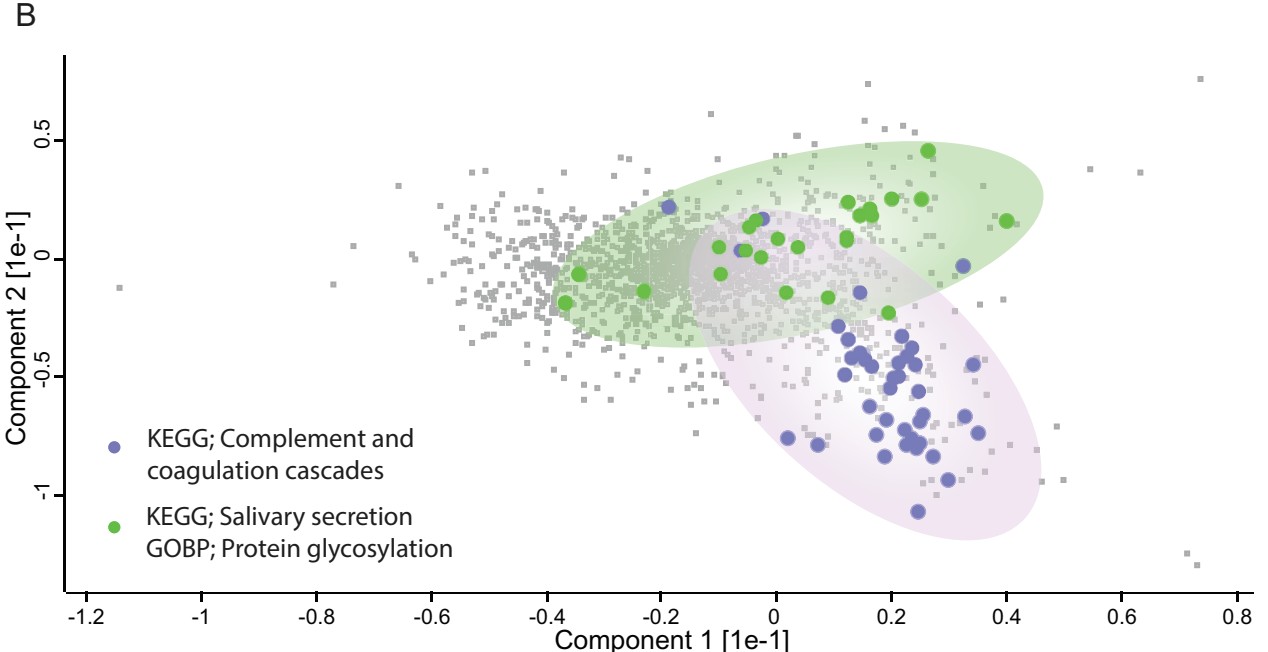

**Figure 2 Principal Component Analysis.** (A) PCA plot of individuals with caries (blue), periodontitis (red) and orally healthy individuals (green). (B) Loadings driving the separation of the PCA plot are mainly proteins belonging to complement and coagulation cascades (purple) for discriminating diseased from healthy individuals. Proteins belonging to salivary secretion and protein glycosylation (green) are mainly defining the healthy individuals.

although three periodontitis individuals cluster together with the healthy group (Fig. 3). We identified three main protein clusters. Cluster I contains human proteins that are higher expressed in both disease groups compared to controls, and 10 of 20 proteins in

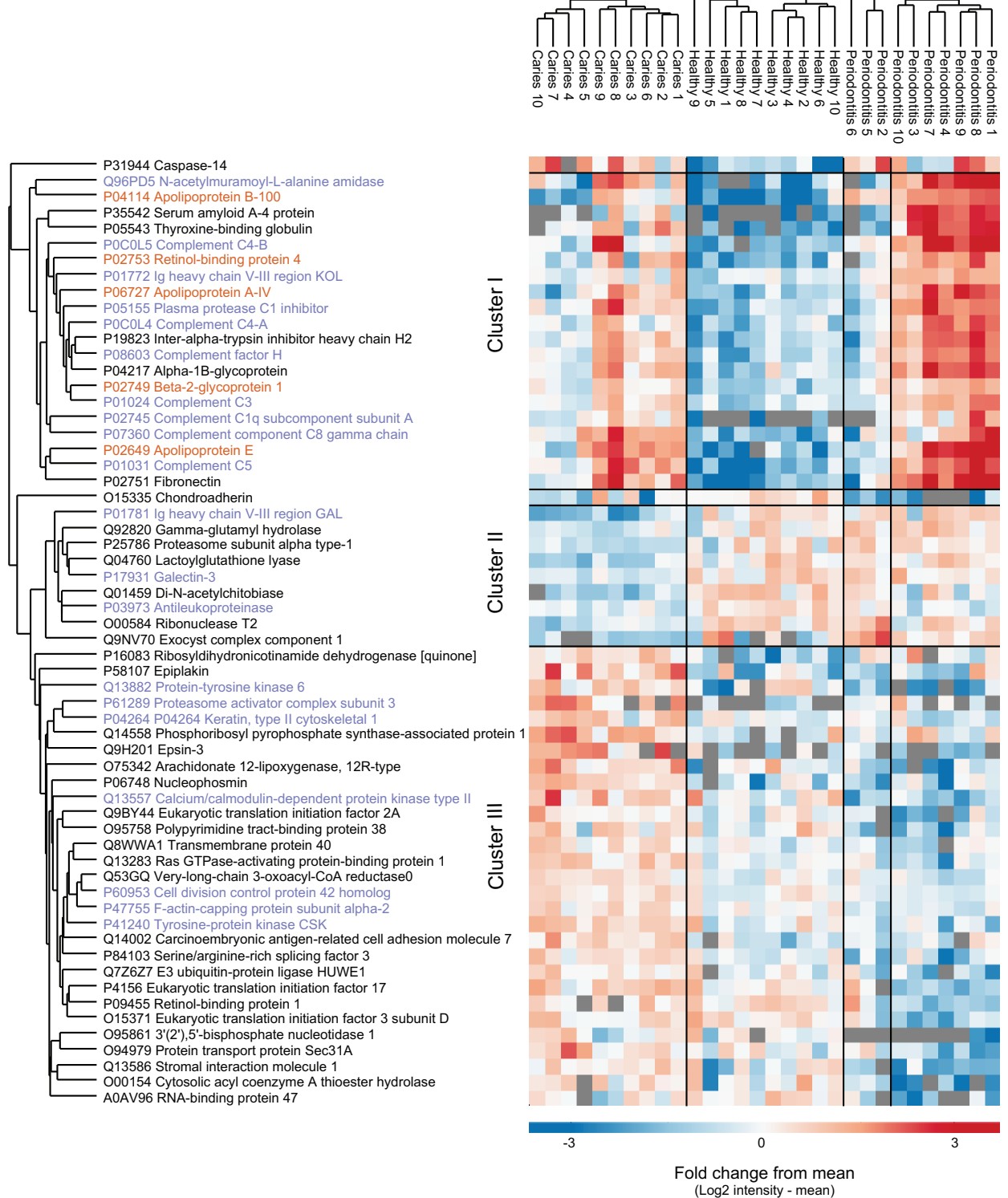

**Figure 3 Potential biomarkers of oral health and disease.** Intensity-based heat-map of proteins significantly differentially expressed between the three groups. Protein names in purple are associated with innate immune response, protein names in orange are associated with lipid transfer.

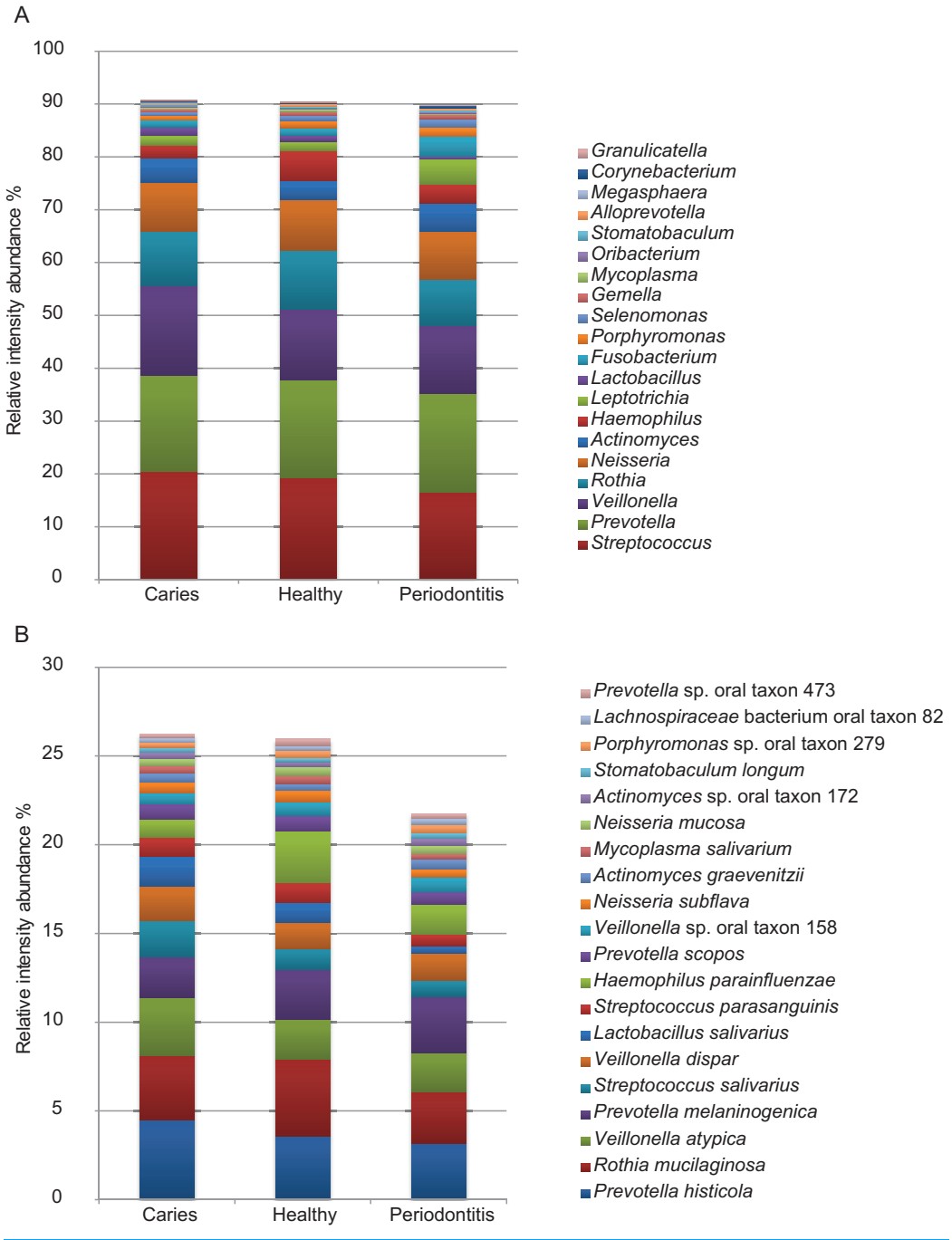

**Figure 4 Predominant bacterial genera and species.** Relative protein intensity abundance of top 20 genera (A) and species (B).

this cluster are associated with the GO term 'innate immune response' (protein name in purple). Cluster II consist of nine proteins that distinguish the individuals with caries from the other groups. In cluster III the protein intensities in the caries group are higher than the mean, for the orally healthy group it is around the mean and for the individuals with periodontitis lower.

## Bacterial protein profiling

Of the 1,946 proteins of bacterial origin identified, approximately 92% and 34% could be assigned to genus and species level, respectively (Table 1). A total of 29 different bacterial genera and 81 species were identified. The five most predominant bacterial genera were *Streptococcus, Prevotella, Veillonella, Rothia* and *Neisseria* collectively representing approx. 70% of the total bacterial mass. The five most predominant bacterial species identified were *Rothia mucilanginosa, Veillonella atypica, Prevotella histicola, Prevotella melaninogenica* and *Streptococcus salivarius*. Abundances of the 20 most predominant bacterial genera and species are displayed in Figs. 4A and 4B. No statistically significant differences were observed between groups at genus or species level. However, at genus level there is a trend of higher proportion of *Veillonella* and lower proportion of *Haemophilus* were associated with dental caries and higher proportions of *Fusobacterium, Leptotrichia* and *Selenomonas* and lower proportions of *Streptococcus, Rothia* and *Haemophilus* were associated with periodontitis, when compared to orally healthy individuals. The same trend is seen at species level where higher proportion of *Veillonella atypica* and lower proportion of *Haemophilus parainfluenzae* were associated with dental caries, and higher proportions of *Fusobacterium periodonticum* and *Leptotrichia wadei* and lower proportions of *Haemophilus parainfluenzae* were associated with periodontitis, when compared to orally healthy individuals. A full list of all bacterial genera and species identified are presented in Table S1.

## KEGG pathway enrichment for bacterial proteins

KEGG pathway enrichment analysis of bacterial proteins resulted in no significant differences with the application of the previously mentioned fold-change and FDR filter criteria. The characterization of functional associations of bacterial proteins is shown in Fig. S2.

## DISCUSSION

The purpose of the present study was to compare metaproteome profiles of saliva from patients with periodontitis or dental caries to that of orally healthy individuals, as we hypothesized that the composition of the salivary metaproteome would associate with oral health status. To the best of our knowledge, this is the first study to characterize both human and bacterial parts of the salivary metaproteome in patients with periodontitis and dental caries.

In this study, proteins of bacterial origin constituted 46% of the proteome diversity, despite only 3% of the total biomass being bacterial. This agrees with the previously reported approx. 1% of DNA in saliva being of bacterial origin (*Lazarevic et al., 2012*).

We identify 1,946 different bacterial proteins, which is in the same range as the pioneering studies of the salivary metaproteome (*Rudney et al., 2010*; *Jagtap et al., 2012*) but substantially higher than in dental plaque (983 proteins) (*Belda-Ferre et al., 2015*). From the total 1,946 bacterial proteins, 92% and 34% could be assigned to genus and species level, respectively (Table 1). The percentage of bacterial proteins identified at genus and species level is considerably higher than what has previously been accomplished in

metaproteomic analysis of saliva (*Rudney et al., 2010*; *Jagtap et al., 2012*). *Streptococcus*, *Prevotella*, *Veillonella*, *Rothia* and *Neisseria* were the most predominant genera identified, constituting approx. 70% of the biomass across all samples (Fig. 4A). This phylogenetic distribution is in line with analysis of the same samples using next-generation sequencing (*Belstrøm et al., 2016b*), and with previous metaproteomic analysis of saliva in oral health (*Rudney et al., 2010*; *Jagtap et al., 2012*; *Grassl et al., 2016*). By contrast, an analysis of 17 plaque samples from patients with dental caries and healthy controls by metagenomics, metatranscriptomics and metaproteomics found different bacterial compositions in dental plaque at DNA, mRNA and protein level (*Belda-Ferre et al., 2015*). This may reflect differences between studying the metabolically active dental plaque biofilm and the planktonic, metabolically inactive state of the salivary microbiota, and it is in concordance with the functional annotation analysis performed (Fig. S2). Moreover, the finding of higher proportions of *Veillonella* in saliva samples from patients with caries and higher proportions of *Fusobacterium* in samples from periodontitis patients confirms findings from 16S analysis of the same samples (*Belstrøm et al., 2016b*). Interestingly, specific oral bacterial species such as *Veillonella parvula* and *Fusobacterium periodonticum* have been reported to associate with dental caries and periodontitis, respectively (*Takahashi & Nyvad, 2011*; *Colombo et al., 2009*).

Furthermore, 2,090 different proteins of human origin were identified, which is more than in metaproteome profiling of dental plaque (*Belda-Ferre et al., 2015*) and less than a recent study that identified more than 3,700 different human proteins in a mouth swab analysis (*Grassl et al., 2016*). The higher number of human protein identifications in mouth swabs is probably due to swabbing the inside of the complete oral cavity including the inside of the cheek. In this study, we used stimulated saliva samples, which may have diluted the concentration of proteins within the samples compared to that of unstimulated saliva (*Yakob et al., 2014*; *Schafer et al., 2014*). This will of course also affect number of identifications. Based on this finding, unstimulated saliva samples may be preferred for in-depth analysis of the salivary proteome. However, as collection of unstimulated saliva samples is considerably more intricate and time-consuming than collection of stimulated saliva samples, the feasibility of using unstimulated saliva samples for population-based biomarker screening approaches may be limited (*Belstrøm et al., 2016b*). In addition, we have recently compared the salivary microbiota in unstimulated and stimulated saliva samples, collected from the same individuals, and reported that comparable microbiotas could be identified using the two types of samples (*Belstrøm et al., 2016a*). Consequently, stimulated saliva samples were used in this study.

Data on the human profile of the salivary metaproteome showed differences between oral health and disease, as proteins involved in innate immunity and inflammatory proteins were more abundantly expressed in saliva samples from patients with periodontitis and dental caries than orally healthy individuals (Fig. 3). Thus, by use of a contemporary metaproteomics approach we were able to explore that salivary expression of proteins from the innate immune system associates with periodontitis and dental caries. Interestingly, these data are in line with previous reports on periodontitis patients

(*Cole et al., 1981*; *Aurer et al., 1999*). Likewise, it has been reported that active components of the complement system in the gingival crevicular fluid associates with both periodontitis (*Schenkein & Genco, 1977*; *Courts et al., 1977*) and gingivitis (*Patters, Niekrash & Lang, 1989*; *Attströum et al., 1975*). Increased local activation of the complement system in the periodontal tissues increases vascular permeability, vasodilatation and recruitment of inflammatory cells, resulting in excessive release of reactive oxygen species, proteolytic enzymes and interleukins (*Okada & Silverman, 1979*; *Okuda & Takazoe, 1980*; *Watanabe et al., 1997*). Furthermore, serum levels of complement proteins, has been suggested to express a linear relationship with the degree of periodontal inflammation (*Henry et al., 1987*). Gingivitis is a mild form of gum disease that results in irritation, redness and swelling caused by inflammation of the gums. Thus, the abundant expression of complement proteins and inflammatory mediators in saliva might reflect either a spillover from the gingival crevicular fluid, or alternatively, mirror increased serum levels of these proteins. Notably, while the complement system has been acknowledged to have a profound role in the pathogenesis of periodontitis (*Damgaard et al., 2015*), the complement system seems to have limited impact on development of dental caries. The expression of complement proteins and other inflammatory proteins in saliva from patients with dental caries is most likely associated with gingivitis in the periodontal tissues adjacent to approximal and gingival caries lesions, and presumably not directly associated with presence of dental caries as such.

## CONCLUSION

Quantitative proteomics data from the present investigation suggest that the salivary microbiota predominantly thrives in a planktonic state with limited metabolic activity, as comparable microbial compositions of the salivary microbiota were obtained based on different omics analysis. Thus, the bacterial part of the metaproteome seems to be inadequate for biomarker analysis of periodontitis and caries. Conversely, a set of human proteins hold the potential to be used as future biomarkers of oral disease status. However, the cross-sectional study design obviously hampers the possibility to address causality of this observation. Thus, future large-scale longitudinal studies of human saliva proteome changes are warranted to reveal the full potential of quantitative proteomics of saliva as a technique to discover biomarkers of oral health and disease.

### Funding

This work was in part funded by the University of Copenhagen (KU2016 programme). Work at Novo Nordisk Foundation Center for Protein Research (CPR) is funded in part by a donation from the Novo Nordisk Foundation (Grant number NNF14CC0001). The funders had no role in study design, data collection and analysis, decision to publish, or preparation of the manuscript.

## Grant Disclosures

The following grant information was disclosed by the authors:
University of Copenhagen: KU2016 programme.
Novo Nordisk Foundation: NNF14CC0001.

## Competing Interests

The authors declare that they have no competing interests.

## Author Contributions

- Daniel Belstrøm conceived and designed the experiments, performed the experiments, analyzed the data, wrote the paper.
- Rosa R. Jersie-Christensen conceived and designed the experiments, performed the experiments, analyzed the data, wrote the paper, prepared figures and/or tables.
- David Lyon analyzed the data, prepared figures and/or tables, reviewed drafts of the paper.
- Christian Damgaard reviewed drafts of the paper.
- Lars J. Jensen contributed reagents/materials/analysis tools, reviewed drafts of the paper.
- Palle Holmstrup contributed reagents/materials/analysis tools, reviewed drafts of the paper.
- Jesper V. Olsen contributed reagents/materials/analysis tools, reviewed drafts of the paper.

## Human Ethics

The following information was supplied relating to ethical approvals (i.e., approving body and any reference numbers):

All participants signed an informed consent prior to participation, and the study was approved by the regional ethical committee (H-15000856-53175) and reported to the Danish Data Authorization (2015-54-0970).

## Data Deposition

ProteomeXchange Pride: https://www.ebi.ac.uk/pride/archive/projects/PXD004319/files.

## Supplemental Information

Supplemental information for this article can be found online at http://dx.doi.org/10.7717/peerj.2433#supplemental-information.

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
