# Peer review of "Metaproteomics of saliva identifies human protein markers specific for individuals with periodontitis and dental caries compared to orally healthy controls"

_PeerJ, doi:10.7717/peerj.2433_

## Round 0.1 · original submission · Minor Revisions

Your manuscript has recieved three reviews. Largely there are minor comments that need to be addressed as you can see below.

Reviewer 1 ·

Basic reporting

No Comments

Experimental design

No Comments

Validity of the findings

No Comments

Additional comments

The paper entitled “Metaproteomics of saliva identifies human protein markers specific for individuals with periodontitis and dental caries compared to orally healthy controls” is an interesting paper that aims to identify biomarkers to distinguish dental caries from periodontitis and healthy subjects. To fulfil this goal, authors have studied stimulated saliva and conjugated with metaproteomics approach. At the end they identified a predominance of some biological processes associated to each one. Despite this, there are some points that should be addressed before final acceptance. For example:
-authors used stimulated saliva arguing that is more versatile for biomarker discovery. But stimulated saliva could lead to the dilution of bacteria content and not reflecting properly the bacteria colonization and their association to the pathologies. This point could somehow justify the conclusion mentioned by authors “Thus, the bacterial part of the metaproteome seems to be inadequate for biomarker”. Have the authors compared some stimulated saliva samples vs non-stimulated saliva samples?

Moreover, authors identified several bacteria such as “Veillonella and lower proportion of Haemophilus were associated with dental caries and higher proportions of Fusobacterium, Leptotrichia and Selenomonas and lower proportions of Streptococcus, Rothia and Haemophilus were associated with periodontitis”. How can authors justify these findings and their association to dental caries and periodontitis? All these findings were not explored in the discussion that should also be taken in consideration

Minor:
There are a predominance of own references and some key references should also be mentioned.

·

Basic reporting

No Comments

Experimental design

No Comments

Validity of the findings

No Comments

Additional comments

The authors used proteomic method to study the composition of saliva and tried to make some meaningful annotations. The Material and Methods appear sound and the paper is mostly well written but I have the following comments/criticism.

Major comments:
Supplementary data not available to judge the quality of the proteomics data.

Minor comments

1: In line 130: it is not clear what is that “(2015)” stands for.

2: In line 171: a “,” should be added before “KEGG”.

3: In line 192: the protein biomass and numbers between groups are not the same (Figure S1). Is “no differences” in this line actually means “no significant differences”. If so, please add description of statistical analyses in materials and methods

4: In line 221, it is unclear how statistics are carried out for the “no statistical differences” here. It would be helpful to have the explanation of statistical analyses in materials and methods.

·

Basic reporting

This articles investigates differences in proteins found in saliva from periodontitis, caries and health. It is very well written with clear presentation of the methods, results and an appropriate discussion. It is well and appropriately referenced, clearly showing how it fits in and adds to the current literature. The structure is good and the figures quite clear.

Experimental design

The experimental design is appropriate for this kind of study. It appears to be thorough and rigorous. They are well described.

Validity of the findings

The findings are valid given the methodology and appropriately analysed and described in the text. The tables and figures clearly show the findings. The discussion is appropriate to the results and methods.

Additional comments

Really, I just have a couple of minor questions:

Although the study details have already been published, I feel some comment about sample size is appropriate.

Do you need to adjust the p value for multiple comparisons?

---

## Round 0.2 · accepted · Accept

Thank you for the changes you have made in response to the reviewers' comments.